# Multi-Target Tracking Algorithm Combined with High-Precision Map

**DOI:** 10.3390/s22239371

**Published:** 2022-12-01

**Authors:** Qingru An, Yawen Cai, Juan Zhu, Sijia Wang, Fengxia Han

**Affiliations:** 1Beijing Rxbit Electronic Technology Co., Ltd., Beijing 100081, China; 2School of Information and Electronics, Beijing Institute of Technology, Beijing 100081, China; 3School of Physics & Electronic Engineering, Hubei University of Arts and Science, Xiangyang 441053, China; 4The Trade Desk Inc., 42 N Chestnut St., Ventura, CA 93001, USA; 5School of Software Engineering, Tongji University, Shanghai 201804, China

**Keywords:** FMCW radar, high-precision map, Kalman filtering, data association

## Abstract

On high-speed roads, there are certain blind areas within the radar coverage, especially when the blind zone is in curved road sections; because the radar does not have the measurement point information in multiple frames, it is easy to have a large deviation between the real trajectory and the filtered trajectory. In this paper, we propose a track prediction method combined with a high-precision map to solve the problem of scattered tracks when the targets are in the blind area. First, the lane centerline is fitted to the upstream and downstream lane edges obtained from the high-precision map in certain steps, and the off-north angle at each fitted point is obtained. Secondly, the normal trajectory is predicted according to the conventional method; for the extrapolated trajectory, the northerly angle of the lane centerline at the current position of the trajectory is obtained, the current coordinate system is converted from the north-east-up (ENU) coordinate system to the vehicle coordinate system, and the lateral velocity of the target is set to zero in the vehicle coordinate system to reduce the error caused by the lateral velocity drag of the target. Finally, the normal trajectory is updated and corrected, and the normal and extrapolated trajectories are managed and reported. The experimental results show that the accuracy and convergence effect of the proposed methods are much better than the traditional methods.

## 1. Introduction

In recent years, the number of cars has been growing, and the pressure on highway operations has risen. The key point of building an intelligent traffic system (ITS) is monitoring road traffic. The most common sensor used for highway monitoring is a vision sensor [1], such as a camera commonly found on highways. Although video image processing technology is a new traffic detection method that has been gradually developed in recent years, it is wireless, flexible in use, can detect multiple traffic parameters at once, and has a large detection range, and it has a very broad application prospect with the rise of high-definition cameras, deep learning, artificial intelligence, and other technologies [2,3,4,5,6,7,8,9,10]. However, the detection performance of vision sensors is significantly reduced under bad weather conditions such as low light, rain, snow, and fog. In addition, vehicle detection on highways using vision sensors is also prone to driver privacy issues. Other researchers have proposed using satellite remote sensing images to identify vehicles and roads to control traffic congestion on roads and identify traffic density quickly and accurately [11,12,13,14,15,16,17,18,19,20,21,22,23,24,25,26,27]. However, this method is computationally intensive and is not conducive to real-time processing. In the development of highways into the process of intelligent construction, millimeter-wave radar has the features of all-day and all-weather operation, is independent of the external light environment, and the radar sensor has the characteristics of high accuracy, small size, and being a good estimate of the location of the vehicle target and speed information, so it has been widely used in the field of high-speed traffic [28].

In recent years, more and more scholars have begun researching the use of radar to monitor the traffic environment, and the problem of multiple object tracking (MOT) [29,30,31,32,33] has been a subject of great importance to scholars both at home and abroad. Different approaches for blind zone tracking have been proposed. A Joint Integrated Probabilistic Data Association Filter (JIPDAF) [34] method is proposed to track pedestrians in front vision and vehicle blind spots. The JIPDAF method is based on a camera and radar sensor, and it is easier to suppress false alarms of blind spot radar (BSR). Researchers have proposed a non-myopic multi-sensor collaborative scheduling method [35] for tracking the ground maneuvering target in the presence of the detection blind zone (DBZ). The method not only considers the immediate reward of the sensor scheduling action, but also takes into account the future expected reward over the prediction time horizon. Most of the existing methods of the blind zone for target detection and tracking are based on various sensors, but few researchers are involved in target tracking in a single radar blind zone on the highway.

There are certain blind zones in the radar coverage, especially when there are radar blind spots on curved roads. In addition, due to the absence of measurement point information in multiple frames, it is easy to have a significant deviation between the real track and the filtered track, resulting in a scattered track. Therefore, to solve this problem, this article proposes a track prediction method combined with high-precision maps. First, the lane centerline is fitted to the upstream and downstream lane edges obtained from the high-precision map in certain steps, and the off-north angle at each fitted point is obtained. Secondly, the normal trajectory is predicted according to the conventional method, and for the extrapolated trajectory, the northerly angle of the lane centerline at the current position of the trajectory is obtained, the current coordinate system is converted from the north-east-up (ENU) coordinate system to the vehicle coordinate system, and the lateral velocity of the target is set to zero in the vehicle coordinate system to reduce the error caused by the lateral velocity drag of the target. Finally, the normal trajectory is updated and corrected, and the normal and extrapolated trajectories are managed and reported.

This paper is organized as follows: the relevant theory is briefly introduced in Section 2. Section 3 provides an overview of the algorithmic system, including the main techniques of the algorithm and the algorithmic process. Experimental results and analysis are given in Section 4. Finally, the conclusions are given in Section 5.

## 2. Relevant Theory

### 2.1. Coordinate System Conversion

In the article, four coordinate systems are mainly involved: WGS-84 geodetic coordinate system, north-east-up (ENU) coordinate system, radar coordinate system, and vehicle coordinate system.

The WGS-84 coordinate system is the abbreviation of the 1984 World Geodetic System (WGS). It represents points on the Earth in terms of three quantities: latitude, longitude, and altitude, also known as the LLA coordinate system. As shown in Figure 1, latitude is the angle between the normal of the reference ellipsoidal plane across point P and the equatorial plane, with latitude values ranging from −90° to +90°, positive for the northern hemisphere and negative for the southern hemisphere. Longitude is the angle between the meridian plane passing through point P and the prime meridian, with longitude values between −180° and +180°. The altitude is the distance normal to the reference ellipsoid from point P. It is negative inside the reference ellipsoid and positive outside.

The Earth-Centered, Earth-Fixed (ECEF) coordinate system is a Cartesian coordinate system with the center of the earth as the origin, as shown in Figure 2. The *z*-axis is parallel to the Earth’s axis and points to the north pole, the *x*-axis points to the intersection of the prime meridian and the equator, and the *y*-axis points to the intersection of 90° east longitude and the equator. Where a represents the major axis of the earth and b represents the minor axis of the earth.

The local Cartesian coordinate system is also called the north-east-up (ENU or NEU) coordinate system, with the selected observation point as the origin, the due east direction as the *x*-axis, the due north direction as the *y*-axis, and the direction pointing to the sky as the *z*-axis.

The radar coordinate system is a Cartesian coordinate system with the radar installation position as the origin, with the radar beam pointing direction as the *y*-axis, and with the *y*-axis oriented 90° clockwise as the *x*-axis.

The vehicle coordinate system is a Cartesian coordinate system with the selected observation point as the origin, with the direction of vehicle travel as the *y*-axis, and with the *y*-axis 90° clockwise as the *x*-axis.

#### 2.1.1. LLA Coordinate System Converted to ENU Coordinate System

The conversion of the LLA coordinate system to the ENU coordinate system requires the use of the ECEF coordinate system as a transition. For a point in space, the conversion relationship from longitude–latitude–altitude coordinates (L,B,H) to the Cartesian coordinate system (X,Y,Z) is as follows.
(1)$$\{\begin{array}{c} X=(N+H)\cos B\cos L\\ Y=(N+H)\cos B\sin L\\ Z=[N(1-e^{2})+H]\sin B \end{array}$$
where $N=a/\sqrt{1-e^{2}\sin^{2}B}$ and N is the radius of curvature in the prime vertical at that point. $e^{2}=(a^{2}-b^{2})/a^{2}$; a, b, and e are the long semi-axis, short semi-axis, and the first eccentricity of the reference ellipsoid corresponding to the ECEF coordinate system, respectively.

The conversion of the ECEF coordinate system (X,Y,Z) to the ENU coordinate system (E,N,U) is calculated as follows.
(2)$$\left\{ \begin{array}{l} E=-\sin L_{0}\cdot \left(X-X_{0}\right)+\cos L_{0}\cdot \left(Y-Y_{0}\right)\\ N=-\sin B_{0}\cos L_{0}\cdot \left(X-X_{0}\right)-\sin L_{0}\sin B_{0}\cdot \left(Y-Y_{0}\right)+\cos B_{0}\cdot \left(Z-Z_{0}\right)\\ U=\cos B_{0}\cos L_{0}\cdot \left(X-X_{0}\right)+\cos B_{0}\sin L_{0}\cdot \left(Y-Y_{0}\right)+\sin B_{0}\cdot \left(Z-Z_{0}\right) \end{array}\right.$$
where $\left(X_{0},Y_{0},Z_{0}\right)$ is the representation of the coordinate origin of the ENU coordinate system in the ECEF coordinate system, and $\left(L_{0},B_{0},H_{0}\right)$ is the representation of the coordinate origin of the ENU coordinate system in the LLA coordinate system.

Equation (2) can also be expressed as
(3)$$\left[\begin{array}{c} E\\ N\\ U \end{array}\right]=R\cdot \left[\begin{array}{c} X-X_{0}\\ Y-Y_{0}\\ Z-Z_{0} \end{array}\right]$$
where $R=\left[-\begin{array}{ccc} -\sin L_{0} & \cos L_{0} & 0\\ \sin B_{0}\cos L_{0} & -\sin B_{0}\sin L_{0} & \cos B_{0}\\ \cos B_{0}\cos L_{0} & \cos B_{0}\sin L_{0} & \sin B_{0} \end{array}\right]$.

#### 2.1.2. LLA Coordinate System Converted to Radar Coordinate System

First, the LLA coordinate system is converted into the ENU coordinate system, the installation position where the current radar is located is used as the origin of the ENU coordinate system, and then the ENU coordinate system is rotated by an angle around the *z*-axis according to the magnitude of the off-north angle of the radar (the angle between the radar irradiation direction and the due north direction) to obtain the radar coordinate system. The conversion of the ENU coordinate system to the radar coordinate system is performed as follows.

Figure 3 shows a schematic diagram of the conversion of the ENU coordinate system to the radar coordinate system, where the conversion formula is
(4)$$\{\begin{array}{c} a_{x1}=a_{x}\cos\psi +b_{y}\sin\psi \\ b_{y1}=b_{y}\cos\psi -a_{x}\sin\psi \end{array}$$

#### 2.1.3. LLA Coordinate System Converted to Vehicle Coordinate System

The process of converting the LLA coordinate system to the vehicle coordinate system is the same as the process of converting the LLA coordinate system to the radar coordinate system, except that the origin of the vehicle coordinate system is the radar installation position with the number 0.

### 2.2. System Model

#### 2.2.1. Equation of State

On high-speed roads, vehicles generally maintain a uniform motion, so we use the constant-velocity (CV) model to describe the motion of vehicle targets. The equation of state is an assumption of the law of motion of the target. For example, assuming that the target moves in a uniform linear motion in a two-dimensional plane, the state $\left(x_{k},y_{k}\right)$ of the target at moment $t_{k}$ under the discrete-time system may be expressed as
(5)$$x_{k}=x_{0}+vt_{k}=x_{0}+v_{x}kT$$
(6)$$y_{k}=y_{0}+v_{y}t_{k}=y_{0}+v_{y}kT$$
where $\left(x_{0},y_{0}\right)$ is the position of the target at the initial moment, $v_{x}$ and $v_{y}$ are the velocities of the target in the *x*-axis and *y*-axis, respectively, and T is the sampling time interval.

Equations (5) and (6) can be expressed in recursive form as
(7)$$x_{k+1}=x_{k}+v_{x}T=x_{k}+{\dot{x}}_{k}T$$
(8)$$y_{k+1}=y_{k}+v_{y}T=y_{k}+{\dot{y}}_{k}T$$

Consider the impossibility of obtaining an accurate model of the target and many unpredictable phenomena. In other words, the target cannot do absolute uniform motion, and its velocity must have some small random fluctuations; for example, the target in the process of uniform speed, the driver or environmental disturbances, etc., can cause unpredictable changes in velocity, and these small changes in velocity can be regarded as process noise to the model. Thus, after the introduction of process noise, Equations (7) and (8) should be expressed as follows
(9)$$x_{k+1}=x_{k}+{\dot{x}}_{k}T+\frac{1}{2}v_{x}T^{2}$$
(10)$$y_{k+1}=y_{k}+{\dot{y}}_{k}T+\frac{1}{2}v_{y}T^{2}$$
where $v_{x}$ and $v_{y}$ denote the random variation in the target *x*-axis and *y*-axis velocities, respectively, and the velocity of the target can be expressed as
(11)$${\dot{x}}_{k+1}={\dot{x}}_{k}+v_{x}T$$
(12)$${\dot{y}}_{k+1}={\dot{y}}_{k}+v_{y}T$$

In the uniform velocity model, the state vector describing the dynamic characteristics of the system is $X(k)={\left[x_{k}\;y_{k}\;{\dot{x}}_{k}\;{\dot{y}}_{k}\right]}^{T}$. Then, Equations (9)–(12) can be expressed in matrix form as
(13)$$\left[\begin{array}{c} x(k+1)\\ y(k+1)\\ \dot{x}(k+1)\\ \dot{y}(k+1) \end{array}\right]=\left[\begin{array}{cccc} 1 & 0 & T & 0\\ 0 & 1 & 0 & T\\ 0 & 0 & 1 & 0\\ 0 & 0 & 0 & 1 \end{array}\right]\left[\begin{array}{c} x(k)\\ y(k)\\ \dot{x}(k)\\ \dot{y}(k) \end{array}\right]+\left[\begin{array}{c} \begin{array}{cc} 0.5T^{2} & 0 \end{array}\\ \begin{array}{cc} 0 & 0.5T^{2} \end{array}\\ \begin{array}{cc} T & 0 \end{array}\\ \begin{array}{cc} 0 & T \end{array} \end{array}\right]\left[\begin{array}{c} v_{x}\\ v_{y} \end{array}\right]$$

That is, the target state equation can be expressed as
(14)$$\begin{array}{ll} X(k+1) & =F(k)X(k)+\Gamma (k)v(k)\\ & =F(k)X(k)+V(k) \end{array}$$
where $v(k)={\left[v_{x},v_{y}\right]}^{T}$ is the process noise vector, $F(k)=\left[\begin{array}{cccc} 1 & 0 & T & 0\\ 0 & 1 & 0 & T\\ 0 & 0 & 1 & 0\\ 0 & 0 & 0 & 1 \end{array}\right]$ is the state transfer matrix of the system, $\Gamma (k)=\left[\begin{array}{c} \begin{array}{cc} 0.5T^{2} & 0 \end{array}\\ \begin{array}{cc} 0 & 0.5T^{2} \end{array}\\ \begin{array}{cc} T & 0 \end{array}\\ \begin{array}{cc} 0 & T \end{array} \end{array}\right]$ is the process noise distribution matrix, and V(k) is a zero-mean, white Gaussian measurement noise sequence with covariance Q(k).

#### 2.2.2. Measurement Equation

The measurement equation is an assumption of the radar measurement process; for linear systems, the measurement equation can be expressed as
(15)Z(k+1)=H(k+1)X(k+1)+W(k+1)
where Z(k+1) is the measurement vector; H(k+1) is the measurement matrix; W(k+1) is a zero-mean, white Gaussian measurement noise sequence with covariance R(k+1), i.e., the measurement noise at different moments is independent of each other, and the process noise sequence is assumed to be independent of the measurement noise sequence and the initial state of the target.

When modeling the target in the two-dimensional plane with uniform motion, the corresponding state variable is $X(k)={\left[x_{k}\;y_{k}\;{\dot{x}}_{k}\;{\dot{y}}_{k}\right]}^{T}$, the measurement vector is $Z(k)={\left[x_{k}\;y_{k}\;{\dot{x}}_{k}\;{\dot{y}}_{k}\right]}^{T}$, and the corresponding measurement matrix is
(16)$$H(k)=\left[\begin{array}{cccc} 1 & 0 & 0 & 0\\ 0 & 1 & 0 & 0\\ 0 & 0 & 1 & 0\\ 0 & 0 & 0 & 1 \end{array}\right]$$

### 2.3. Filtering Model

The Kalman filter is a filter under the linear mean square error criterion for non-smooth problems with finite observation intervals.

The minimum mean square error estimate for the time-varying case can be defined as
(17)$$\hat{X}(k|k)=E[X(k)|Z^{k}]$$
where
(18)$$Z^{k}=\{Z(j),j=1,2,\ldots ,k\}$$

The state error covariance matrix accompanying Equation (17) is defined as
(19)$$P(k|k)=E\{[X(k)-\hat{X}(k|k)]{\left[X(k)-\hat{X}(k|k)\right]}^{T}|Z^{k}\}$$

Applying the expectation operator conditioned on $Z^{k}$ to Equation (14) yields a one-step prediction of the state as
(20)$$\begin{array}{ll} \hat{X}(k+1|k) & =E[X(k+1)|Z^{k}]=E[F(k)X(k)+V(k)|Z^{k}]\\ & =F(k)\hat{X}(k|k) \end{array}$$

The error of the predicted value is
(21)$$\tilde{X}(k+1|k)=X(k+1)-\hat{X}(k+1|k)=F(k)\tilde{X}(k|k)+V(k)$$

The one-step prediction covariance is
(22)$$\begin{array}{ll} P(k+1|k) & =E[\tilde{X}(k+1|k){\tilde{X}}^{T}(k+1|k)|Z^{k}]\\ & =E\{[F(k)\tilde{X}(k|k)+V(k)][F^{T}(k){\tilde{X}}^{T}(k|k)+V^{T}(k)]|Z^{k}\}\\ & =F(k)P(k|k)F^{T}(k)+Q(k) \end{array}$$

Similarly, the predicted value of the measurement can be expressed as
(23)$$\begin{array}{ll} \hat{Z}(k+1|k) & =E[Z(k+1)|Z^{k}]=E\{[H(k+1)X(k+1)+W(k+1)]|Z^{k}\}\\ & =H(k+1)\hat{X}(k+1|k) \end{array}$$

In turn, the difference between the predicted and measured values of the measurement can be found as
(24)$$\tilde{Z}(k+1|k)=Z(k+1)-\hat{Z}(k+1|k)=H(k+1)\tilde{X}(k+1|k)+W(k+1)$$

The predicted covariance (or new interest covariance) of the measure is
(25)$$\begin{array}{ll} S(k+1) & =E[\tilde{Z}(k+1|k){\tilde{Z}}^{T}(k+1|k)|Z^{k}]\\ & =E\{[H(k+1)\tilde{X}(k+1|k)+W(k+1)][H^{T}(k+1){\tilde{X}}^{T}(k+1|k)+W^{T}(k+1)]|Z^{k}\}\\ & =H(k+1)P(k+1|k)H^{T}(k+1)+R(k+1) \end{array}$$

The Kalman gain is
(26)$$K(k+1)=P(k+1|k)H^{T}(k+1)S^{-1}(k+1)$$

Further, the state update equation at moment k+1 can be found as
(27)$$\hat{X}(k+1|k+1)=\hat{X}(k+1|k)+K(k+1)v(k+1)$$
where v(k+1) is the measurement residual, i.e.,
(28)$$v(k+1)=\tilde{Z}(k+1|k)=Z(k+1)-\hat{Z}(k+1|k)$$

The covariance update equation is
(29)$$\begin{array}{ll} P(k+1|k+1) & =P(k+1|k)-P(k+1|k)H^{T}(k+1)S^{-1}(k+1)H(k+1)P(k+1|k)\\ & =[I-K(k+1)H(k+1)]P(k+1|k) \end{array}$$

## 3. System Overview

### 3.1. Main Technologies

#### 3.1.1. High-Precision Map Lane Line Fitting

In the radar Cartesian coordinate system, two lane centerlines are fitted according to the Y values of the road edges given by the high-precision map in certain steps (e.g., 10 m), but the Y values are discrete and the X values corresponding to the Y values selected by a certain step distance each time do not necessarily exist in the high-precision map, so we need to interpolate to obtain the corresponding X values. The Lagrangian interpolation method can find a polynomial that takes exactly the observed value at each observed point, and the details of the algorithm are as follows.

For a certain polynomial function, it is known that there are k+1 given taking points: $\left(x_{0},y_{0}),(x_{1},y_{1}),\ldots ,(x_{k},y_{k}\right)$, where $x_{j}(j=0,1,2,\ldots ,k)$ corresponds to the position of the independent variable, and $y_{j}(j=0,1,2,\ldots ,k)$ corresponds to the value of the function taken at this position.

Assuming that any two distinct $x_{j}$ are different from each other, the Lagrangian interpolation polynomial obtained by applying the Lagrangian interpolation formula is
(30)$$L(x)=\sum_{j=0}^{k}y_{j}l_{j}(x)$$
where each $l_{j}(x)$ is a Lagrangian fundamental polynomial (or interpolating basis function) with the expressions
(31)$$l_{j}(x)=\prod_{i=0,i\neq j}^{k}\frac{x-x_{i}}{x_{j}-x_{i}}=\frac{x-x_{0}}{x_{j}-x_{0}}\ldots \frac{x-x_{j-1}}{x_{j}-x_{j-1}}\frac{x-x_{j+1}}{x_{j}-x_{j+1}}\ldots \frac{x-x_{k}}{x_{j}-x_{k}}$$

The Lagrangian fundamental polynomial $l_{j}(x)$ is characterized by taking the value 1 on $x_{j}$ and 0 at other points $x_{i}(i\neq j)$.

In the actual Lagrangian interpolation process, we use k=2 and select three actual high-precision map lane information points.

#### 3.1.2. Data Association Algorithm

In the data association process, we use the computationally simple Nearest-Neighbor Data Association (NNDA) Algorithm, which works by first setting up a tracking wave gate, and the echoes obtained by the initial screening of the tracking wave gate become candidate echoes.

A tracking wave gate is a subinterval in the tracking space centered at the predicted position of the tracked target. The size of the tracking wave gate should be designed to ensure that the correct echoes are received with a certain probability. The simplest method of correlation wave gate formation is to define a rectangular region in the tracking space, i.e., a rectangular wave gate.

Let the measurement residual v(k+1), the i-th component of the measure Z(k+1), and the predicted value of the measure $\hat{Z}(k+1|k)$ be denoted by $v_{i}(k+1)$, $Z_{i}(k+1)$, and ${\hat{Z}}_{i}(k+1|k)$, respectively, and the *i*-th row and *j*-th column element of the predicted covariance S(k+1) be denoted by $S_{ij}$. When all components of the measure Z(k+1) satisfy Equation (32), the measure Z(k+1) is said to fall into the rectangular wave gate and the measure is a candidate echo.
(32)$$|v_{i}(k+1)|=|Z_{i}(k+1)-{\hat{Z}}_{i}(k+1|k)|\leq K_{G}\sqrt{S_{ii}}\;\;i=1,2,3,4$$
where $K_{G}$ is the wave gate constant.

In practice, we first set a smaller wave gate ($K_{G}=2$) and initially select the association list of the track and the measurement; the association list includes the index of the track and the measurement points; the correlation weights and the association weights are calculated as shown in Equation (33); the square of the Mahalanobis distance between the track and the measurement points is used as the weights.

The remaining unassociated tracks are associated with the measurement points for a second large wave gate ($K_{G}=5$), and to prevent the large wave gate association weights from being smaller than the association weights of the small wave gate, the association weights of the large wave gate are all increased by a larger fixed value, and the associated results are merged into the association list. Finally, the optimal association is obtained from the association list to obtain the final association list according to the idea of the greedy algorithm.
(33)$$d^{2}(Z)={\left[Z-\hat{Z}(k+1|k)\right]}^{T}S^{-1}(k+1)[Z-\hat{Z}(k+1|k)]$$

#### 3.1.3. Track Prediction Combined with High-Precision Map

When the vehicle target is located in the radar blind zone, continuous multi-frame prediction of the track tracking process is needed as the track filtering result when there is no measurement point, especially when the blind zone is at a curve. The filtering result without measurement is likely to make the filtered track deviate greatly from the real track, resulting in the track divergence. Therefore, instead of using the velocity vector of the track itself for position prediction within the radar blind spot, the coordinates of the track position and velocity information are converted from the ENU coordinate system to the vehicle coordinate system by matching the off-north angle of the lane centerline. In the vehicle coordinate system, the lateral velocity of the target becomes zero, and the target is predicted according to the lane direction, which can prevent the divergence of the track due to its lateral velocity component. After completing the track filtering, the vehicle coordinate system is transferred back to the ENU coordinate system.

### 3.2. Tracking Master Process

On the highway, the tracking of vehicle targets requires the measurement information of the target to correct the predicted state of the target. However, when the target is in the radar blind zone, especially in the curved road, as shown in Figure 4, the predicted track with no measurement point correction in multiple frames will be output as the final track, which is likely to cause the output track to deviate from the real track or even diverge. To solve this problem, when the target is driving in a blind zone or the measurement is missed, the track can be predicted by combining high-precision maps. The algorithm steps are as follows:

Step 1: Divide the four edges of the high-precision map according to the coverage area of radar power.

Step 2: Using the radar’s latitude and longitude coordinates and off-north angle information given by the high-precision map, the four sidelines within each radar coverage area are converted from the LLA coordinate system to the corresponding radar coordinate system under the corresponding radar.

Step 3: Using the road edge data in the radar coordinate system, according to the method in Section 3.1.1, the road edge is fitted with a certain step (such as 10 m) to obtain the corresponding horizontal coordinate values, and then the coordinates of the road centerline at the corresponding distance are obtained.

Step 4: By taking the position relationship between the centerline points of two adjacent lanes, the offset angle Δθ of each differential point in the radar coordinate system is obtained, and then, combined with the current radar off-north angle θ, an angle $\theta^{\prime }=\theta +\Delta \theta$ can be calculated as the off-north angle of the lane of the differential section.

Step 5: Traverse the list of tracks reported at the last moment in the ENU coordinate system, and predict the position and speed information of each track.

Step 6: Adaptively set the predictive model noise Q for the transient track list according to the variable tick values and update the covariance matrix P of the track according to Equation (22).

Parameter tick is mainly used to set the track reporting module and the covariance matrix Q of noise. In the experiment, when tick>8, the track is considered to be a stable track, reaching the standard of track reporting. For the covariance matrix Q of noise, set Q=k⋅diag[0.0009 0.0009 0.0004 0.0004] during initialization, where k is the coefficient.

When tick≤8, set k=1, and it is assumed that the mobility of the target movement is small, which is beneficial to filtering the unstable track. When tick>8, setting k=100 can effectively track the target with strong mobility. The settings of the above parameters are all obtained after the comparative analysis of the experimental results, and they have obvious advantages in the experimental results.

Step 7: Associate the trajectory with the measurement points once with a smaller wave gate ($K_{G}=2$) to obtain a list of all associations, which includes the indexes and association weights of the tracks and the measurement points.

Step 8: Associate the remaining unassociated tracks with the measurement points for a second time with a large wave gate, and merge the association results into the association list.

Step 9: Obtain the optimal association from the association list to obtain the final association list according to the greedy algorithm.

Step 10: Update the track states according to Equations (23)–(29).

Step 11: For the track without measuring point information, the prediction result of Step 5 is canceled, the track prediction method combined with a high-precision map is adopted, and add 1 to the track quality parameter age. First, the track is transferred from the ENU coordinate system to the radar coordinate system according to Equation (4), and then the target track is transferred from the ENU coordinate system to the vehicle coordinate system according to the Y value of the track matching the off-north angle of the corresponding lane center line. The lateral velocity component of the target is set to zero, and the predicted track is predicted according to the vehicle motion model. Finally, the predicted track is converted back to the ENU coordinate system.

Parameter age is mainly used for the track termination module and track extrapolation judgment. When age>600, the track termination condition is met. When 5≤age≤600, that is, there are more than 5 frames of the track that are not related to the measurement point, which meets the conditions of track extrapolation. The track filtering algorithm combined with the high-precision map proposed in this paper is adopted. When age>600 is satisfied or there are measurement points on the track for more than 5 consecutive frames, stop track extrapolation and set age to zero.

The track quality parameter age and tick are two criteria to measure the track quality in track management, which are related to whether the target track can be related to the measurement points. If there are no measurement points in the current track, the age value will be increased by 1, and if there are measurement points in the track, the tick value will be increased by 1.

Step 12: Initial track generation for measurement points that are not associated.

Step 13: For the updated list of transient tracks, report the tracks that meet the reporting conditions.

Figure 5 shows the overall flow of the tracking algorithm in this paper. Therefore, when the target is within the radar irradiation range, the traditional track filtering method is used to filter the target track. When the target is in the radar blind area, the multi-target tracking algorithm combined with the high-precision map proposed in this paper is used to filter the target. Compared with the traditional multi-target tracking algorithm, we consider the situation of target tracking in the radar blind zone.

## 4. Results

This section aims to evaluate the track filtering results of the proposed method for targets in radar blind zones in complex scenarios. We compare the filtering results of a radar deployed on a highway in two scenarios: on a straight road and a curved road.

### 4.1. Parameters

The traffic radar used in this experiment is based on a 79 GHz millimeter-wave radar, cascaded with a long-range antenna, a medium-range antenna, and a short-range antenna, each with a different downward tilting installation angle, to achieve a long-range radar coverage of 45 m~550 m, breaking the limitation of the traditional antenna beam coverage and achieving a longer-range radar coverage to meet the current road detection requirements. The specific parameters of the traffic radar are shown in Table 1.

The radar is mounted in the middle of the green belt between the upstream lane and downstream lane at a height of 10 m. Figure 6 depicts the coverage of traffic radar, from which it can be seen that there is a 45 m radar blind zone for a single radar. On high-speed roads, as in Figure 7, two radars are mounted at one radar installation point with opposite irradiation directions, so there is a radar blind zone of approximately 90 m. The supporting components have no shielding surface, which will not interfere with the electromagnetic waves emitted by radar, and will not affect the detection of objects.

### 4.2. Straight Roads

In a straight road, the radar covers five upstream lanes and five downstream lanes. Figure 8 represents the filtered output of the track, which shows that there is a blind zone at the location of the radar installation.

Figure 9 shows the radar track filtering results. Figure 9a is the track filtering result of the traditional method. It can be seen from the figure that although there is no track deviation of the vehicle target in the radar blind zone, the track filtering at the junction of the radar blind zone and the radar irradiation area is not smooth, so it can be concluded that the track filtering result obtained by the traditional method in the radar blind zone is quite different from the real track. Compared with the track filtering results in Figure 9a, the track filtering results in Figure 9b combined with high-precision maps are smoother, and the track filtering results in the radar blind area are closer to the real track of the target. There are some false detection targets in lane 4 and lane 5 of the downstream lane in Figure 9, which belong to the signal detection part. However, this paper discusses the multi-target tracking processing, which belongs to the data processing part, and does not deal with the false detection generated by the signal processing part.

In the experiment, the data we used for the test contained thousands of frames of radar echo data, so what we presented on the result chart was only a small part of the test data. In Figure 9b, the five lanes above the radar are upstream lanes, and the downstream lanes of the five lanes below the radar have opposite driving directions. The red track of the fourth upstream lane and the blue track of the fourth downstream lane are the beginning of the data we selected, but the track of the target before that has not been selected by us. The blue track of the fourth upstream lane and the red track of the third downstream lane are the last stages of our data selection, after which the target track has not been selected. Therefore, the discontinuous tracks in the figure are not the same target, only because some target tracks are not completely presented in the process of intercepting data.

### 4.3. Curved Roads

In the curved road, the radar covers five upstream lanes and five downstream lanes. Figure 10 represents the filtering results of the output trajectories. Track termination due to lack of measurement point information in the blind zone and the extent of the radar’s blind zone can be seen in the figure.

Figure 11 shows the radar trajectory filtering results. Figure 11a shows the results of the conventional track filtering, from which it can be seen that there are two lanes in the radar blind zone where the trajectories deviate from the current lane, and once the target is out of the radar blind zone, the trajectories that deviate from the current lane continue to be filtered out normally due to the availability of measurement point information to correct for the predicted trajectories. If the target trajectories deviate too much from the lane in the radar blind zone, the trajectories will likely not be correlated with the measurement points information at the junction of the radar blind zone and the radar illuminated zone, resulting in tracks termination and restart. Figure 11b shows the results of the trajectory filtering in radar blind zones combined with high-precision maps. It can be seen from the figure that the trajectory prediction method combined with the high-precision map in the radar blind zone can solve the problem of the trajectory deviating from the current lane very well so that even if there is no measurement point information in multiple frames in the radar blind zone, the trajectory can be filtered out well, and there is no problem where the trajectory and the measurement point are not associated at the junction of the radar blind zone and the radar illumination area.

## 5. Discussion

This paper describes a tracking algorithm combined with a high-precision map in detail. It solves the problem that the track filtering result deviates from the lane in the blind area of radar because there is no measurement point information in multiple frames. The smoothness of the track at the junction of the radar blind area and the radar irradiation area is improved.

In this paper, the multi-target tracking algorithm combined with a high-precision map has been verified in two scenes: a straight road and a curved road. The two selected radars are installed in opposite directions, and the longest distance detected by a single radar can reach 550 m. The blind zone of the two radars is about 90 m, and each radar covers five upstream lanes and five downstream lanes.

In the straight road scene, the road in the radar blind zone is straight, and the driving direction of the target is unchanged. Therefore, in the straight road scene, it is rare that the target track deviates from the lane, but the track connection at the junction of the radar blind zone and the radar irradiation area may not be smooth. Compared with the traditional algorithm, the tracking algorithm proposed in this paper can make the track smoother in a straight road scene.

In the curved road scene, the driving direction of the target will change with the road direction. However, there is no measurement information in the radar blind area, so the traditional tracking filtering method cannot track the target effectively, and it is easy for the target track to deviate from the current lane line. Compared with the traditional algorithm, the tracking algorithm proposed in this paper can reduce the drag effect on the lateral distance of the target by changing the coordinate system in the course of track prediction. Therefore, the target track can be accurately tracked in the curved road scene, the situation that the track deviates from the lane line is reduced, and the track is smoother.

Compared with previous studies, the novelty of this paper lies in establishing a complete set of target-tracking algorithms in traffic scenes. Considering the problem of tracking the target in the blind area of radar, the target is tracked and filtered by combining the method of a high-precision map. Compared with the existing methods, this method can effectively track the targets in more complex road scenes and improve the smoothness of the filtered track.

## Figures and Tables

**Figure 1 sensors-22-09371-f001:**
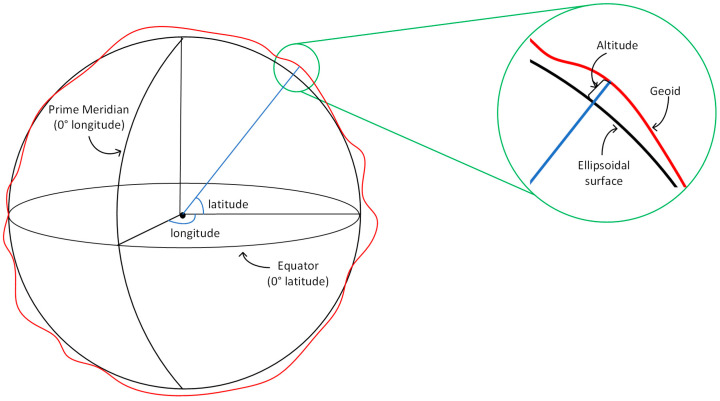
Schematic diagram of LLA coordinate system.

**Figure 2 sensors-22-09371-f002:**
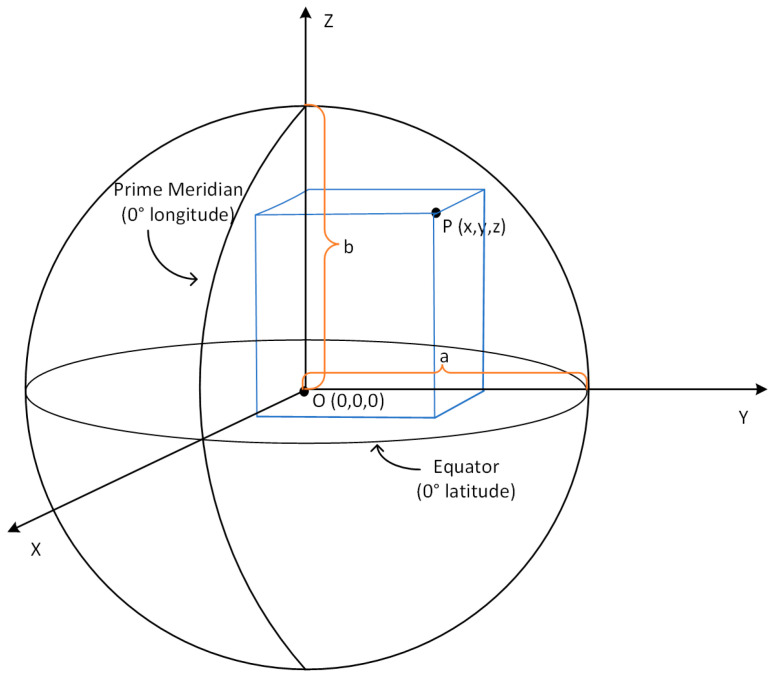
Schematic diagram of ECEF coordinate system.

**Figure 3 sensors-22-09371-f003:**
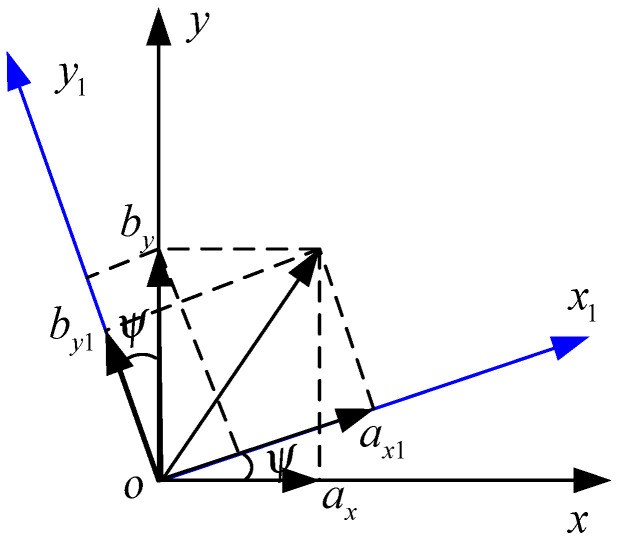
Schematic diagram of conversion from ENU coordinate system to radar coordinate system.

**Figure 4 sensors-22-09371-f004:**
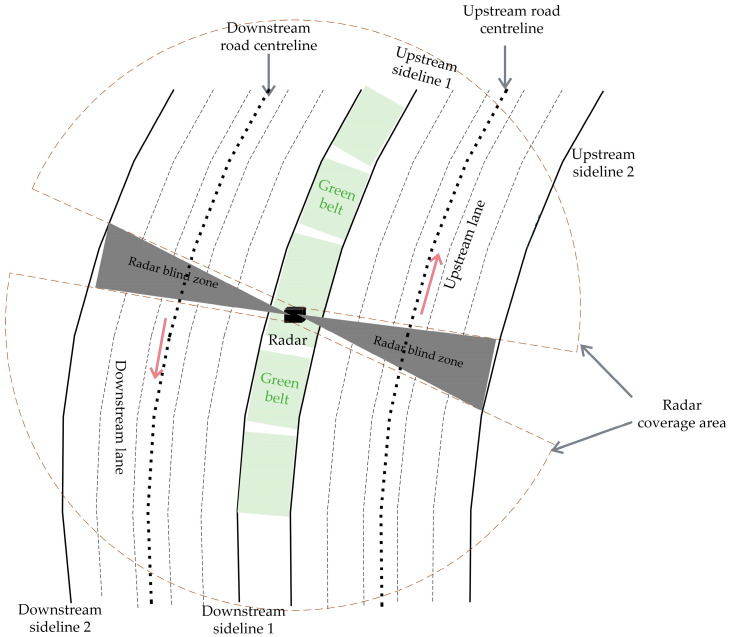
Radar blind zone on a curved road.

**Figure 5 sensors-22-09371-f005:**
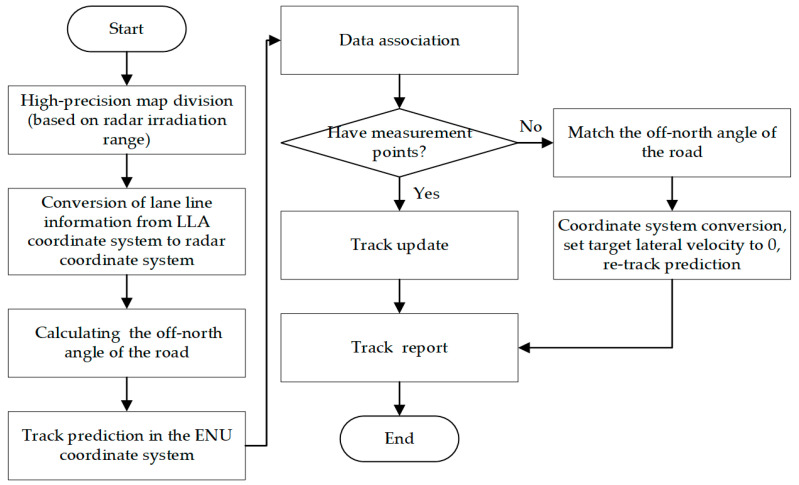
Combined with the high-precision map tracking algorithm.

**Figure 6 sensors-22-09371-f006:**
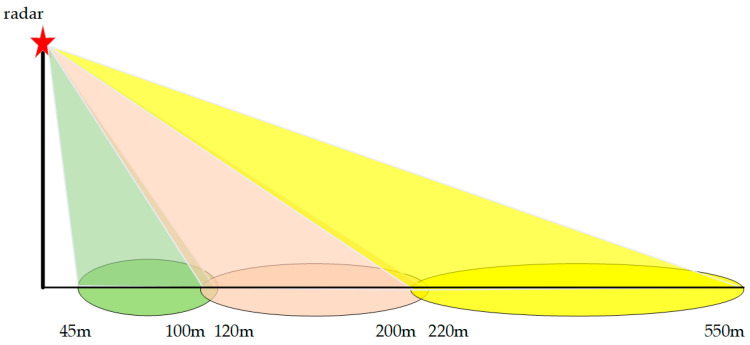
Coverage of traffic radar.

**Figure 7 sensors-22-09371-f007:**
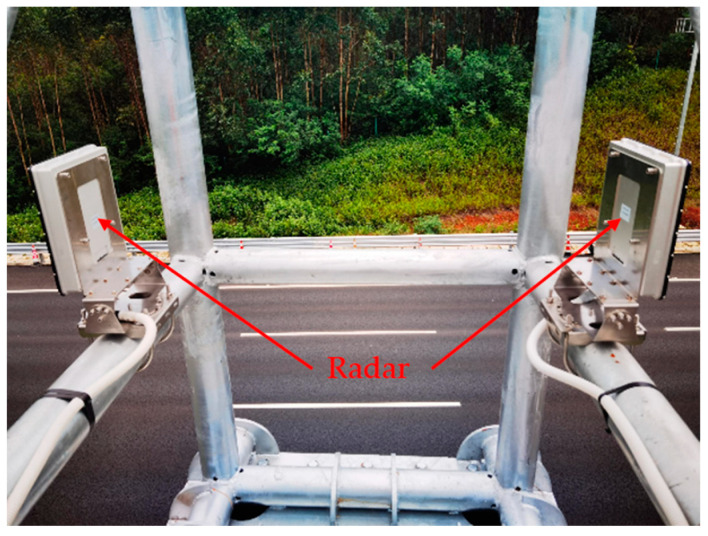
Radar installation in the actual scene.

**Figure 8 sensors-22-09371-f008:**
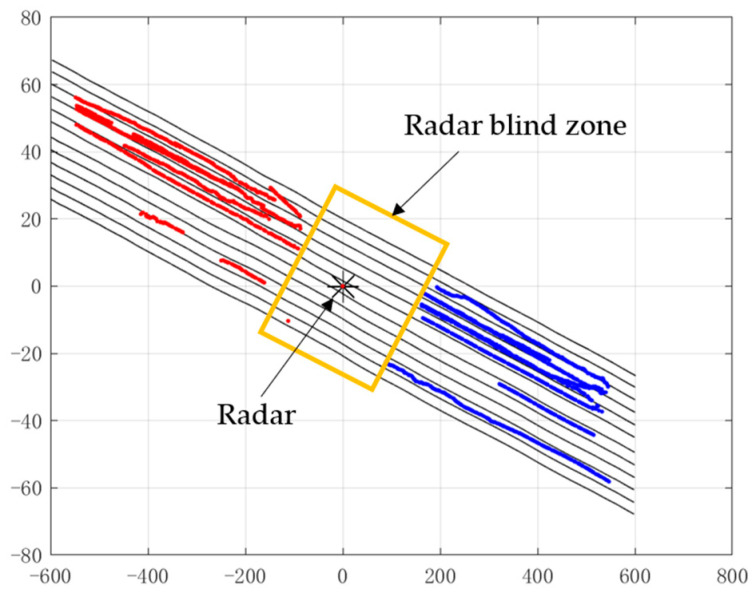
Radar blind zone on a straight road.

**Figure 9 sensors-22-09371-f009:**
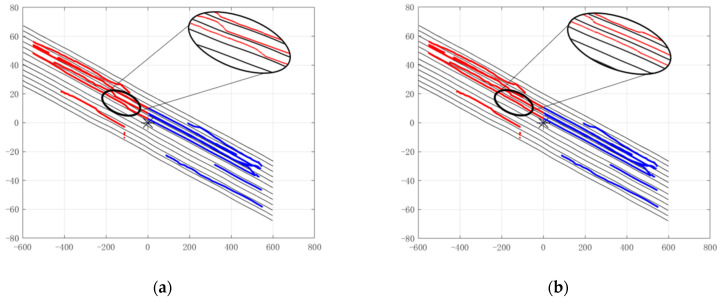
Radar track filtering results on the straight road. (**a**) Traditional method filtering results; (**b**) track filtering results combined with high-precision map.

**Figure 10 sensors-22-09371-f010:**
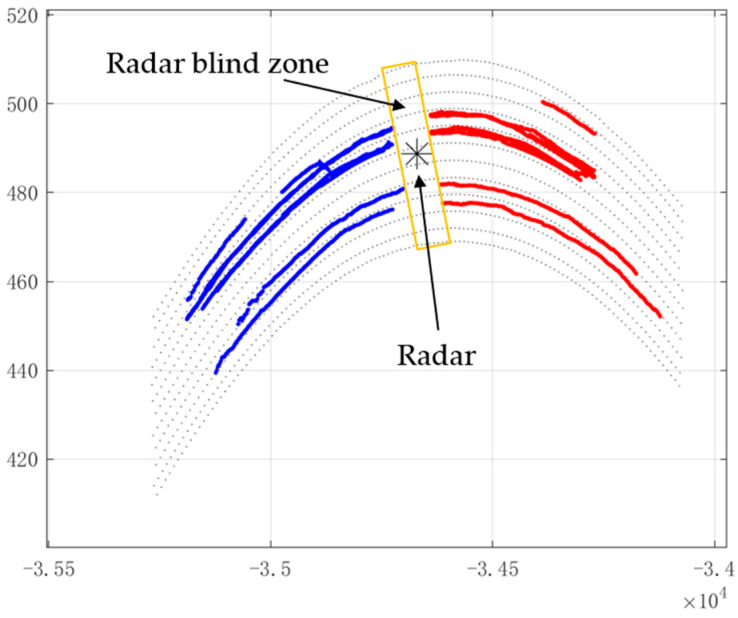
Radar blind zone on a curved road.

**Figure 11 sensors-22-09371-f011:**
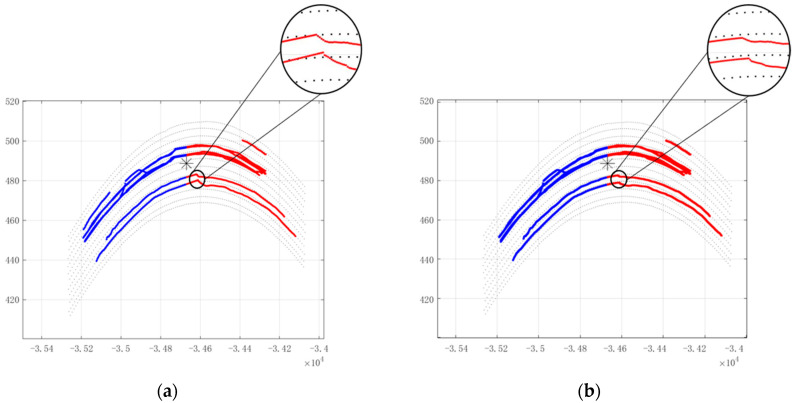
Radar track filtering results on the curved road. (**a**) Traditional method filtering results; (**b**) track filtering results combined with high-precision map.

**Table 1 sensors-22-09371-t001:** Traffic radar parameters.

Parameters	Short-Range	Medium-Range	Long-Range
Installation inclination (elevation angle)	1°	3°	6°
Installation inclination (azimuth angle)	0°	0°	0°
Range	45 m–120 m	100 m–220 m	200 m–550 m
Range resolution	±0.2 m		
FOV (elevation angle)	±6°	±4.4°	±4.4°
FOV (azimuth angle)	Covering 10 lanes (50 m)		
Maximum velocity	−250 km/h ~ +250 km/h		
Velocity resolution	±0.1 m/s		
Installation height	10 m		
Central Frequency	79 GHz		

## Data Availability

Not applicable.

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
