# Peer review of "Multi-Target Tracking Algorithm Combined with High-Precision Map"

_sensors, 2022, doi:10.3390/s22239371_

Round 1

Reviewer 1 Report

This manuscript (paper ID: sensors-2025241) has completed substantial research work and has certain engineering value.  The tracking main process is established and the relevant results are analyzed. Based on the existing presentation, the following aspects need to be improved:

1. Line 202, there are two random characters "&" in the expression of X (k). Please make sure to edit the format of variables and formulas. This problem also occurs many times in other places of the text, such as equation (21, 22).

2. For Figure 7. Radar installation in the actual scene. Whether the support components will affect the detection wave, and how to avoid interference?

3. Please keep the alignment of the pictures consistent, as shown in Figure 10 and Figure 11.

Reviewer 2 Report

The main reason for my decision is the fact that the improvement in the quality of the trajectories seems insignificant. A bigger problem  not addressed by the Authors  are the trajectories in Figure 9 that were not properly connected, viz. the trajectory on the second lane from the top and the trajectory on the second and third lane from the bottom; this shows that the algorithm is not able to connect trajectories if a vehicle changes lanes in the blind zone. Moreover, there seem to be some false detections on the fifth lane from the bottom - what about them? Presented results are not convincing while the important phenomena are ignored.

Moreover, there are some other flaws:

1. Many equations in sections 2.2 and 2.3 are obscured by "&" and "%" symbols.

2. The description of the algorithm in section 3.2 is not clear and should be rewritten in order to enable potential reader to implement it. Is this algorithm used always or only when the target is in the blind area? References to previously introduced equations should be made when relevant. What are the “tick” and “age” parameters? They were not introduced and explained earlier.

Round 2

Reviewer 2 Report

I appreciate the answers to my comments, but there are still two issues that need to be addressed before publication:

1. The explanations – provided in the Author’s reply – should be included in the manuscript (probably in Section 4) because they help the reader understand the presented results.

2. The description of the algorithm in section 3.2 is still not clear – what is the purpose of the “tick” and “age” parameters? For example, “tick” argument is mentioned for the first time in step 6 in the following context: “Adaptive setting of the predictive model noise Q for the transient track list according to the variable tick values…”. How does the value of that parameter influence the setting of Q? There is no information about that. Similar explanation should be added for the “age” parameter.
